# 3D Natural Mesoporous Biosilica-Embedded Polysulfone Made Ultrafiltration Membranes for Application in Separation Technology

**DOI:** 10.3390/polym14091750

**Published:** 2022-04-26

**Authors:** Murali Krishna Paidi, Veerababu Polisetti, Krishnaiah Damarla, Puyam Sobhindro Singh, Subir Kumar Mandal, Paramita Ray

**Affiliations:** 1CSIR-Central Salt and Marine Chemicals Research Institute, Council of Scientific and Industrial Research (CSIR), GB Marg, Bhavnagar 364002, India; muralikrishna.p145@gmail.com (M.K.P.); damarlakittu@gmail.com (K.D.); puyam@csmcri.res.in (P.S.S.); 2Academy of Scientific and Innovative Research (AcSIR), Ghaziabad 201002, India; 3Department of Fibre and Polymer Technology, KTH Royal Institute of Technology, SE-100 44 Stockholm, Sweden

**Keywords:** diatom, biosilica, polysulfone, phase inversion, ultrafiltration, oil separation

## Abstract

Diatoms are the most abundant photosynthetic microalgae found in all aquatic habitats. In the extant study, the spent biomass (after lipid extraction) of the centric marine diatom *Thalassiosira lundiana* CSIRCSMCRI 001 was subjected to acid digestion for the extraction of micro composite inorganic biosilica. Then, the resulting three-dimensional mesoporous biosilica material (diatomite) was used as a filler in polysulfone (PSF) membrane preparation by phase inversion. The fabricated PSF/diatomite composite membranes were characterized by SEM-EDX, TGA, and ATR-IR, and their performances were evaluated. The number of pores and pore size were increased on the membrane surface with increased diatomite in the composite membranes as compared to the control. The diatomite composite membranes had high hydrophilicity and thermal stability, lower surface roughness, and excellent water permeability. Membranes with high % diatomite, i.e., PSF/Dia_0.5_, had a maximum water flux of 806.8 LMH (Liter/m^2^/h) at 20 psi operating pressure. High-diatomite content membranes also exhibited the highest rejection of BSA protein (98.5%) and rhodamine 6G (94.8%). Similarly, in biomedical rejection tests, the PSF/Dia_0.5_ membrane exhibited a maximum rejection of ampicillin (75.84%) and neomycin (85.88%) at 20 Psi pressure. In conclusion, the mesoporous inorganic biosilica material was extracted from spent biomass of diatom and successfully used in filtration techniques. The results of this study could enhance the application of natural biogenic porous silica materials in wastewater treatment for water recycling.

## 1. Introduction

Membrane separation processes occupy a stable position in the treatment of contaminated water. Ultrafiltration (UF) is unique among membrane processes, with diversified applications in various fields, such as protein filtration, bacterial removal, fractionating food products, oil–water separation, etc. [1]. UF is also used as a pretreatment process ahead of the reverse osmosis (RO) in water treatment. In general, UF membranes are made of different hydrophobic polymers, such as polyvinylidene fluoride (PVDF), polysulfone (PSF), polyethersulfone (PES), and polyacrylonitrile (PAN) [2,3]. The hydrophobic nature of these polymers leads to membrane fouling by different organic pollutants. The pollutants are deposited either on the membrane surface or adsorbed into the pores of the membrane, resulting in poor performance of the membranes, which in turn restricts their practical applications. To vanquish the above issue, researchers have explored the use of different inorganic nanoparticles in membranes to improve membrane hydrophilicity and enhance productivity with extended filtration capacity and antifouling properties. For example, membrane fluxes were significantly increased with the addition of nanomaterials such as silicon dioxide (SiO_2_) [4], titanium dioxide (TiO_2_) [5], zinc oxide (ZnO) [6], single- or multiwalled carbon nanotubes (SWCNTs/MWCNTs) [7], silver (Ag) [8], zirconia (ZrO_2_) [9], alumina (Al_2_O_3_) [10], and graphene oxide (GO) [11]. Many inorganic materials are toxic to human health and the environment [12,13]. Conventional nanomaterials have lower porosity and tend to be leached out during long-term filtration. Scientists have reported that non-porous nanoparticles migrate to the membrane surface during the preparation of the membrane by the phase inversion method [14,15]. Currently, there is a demand for such material with uniform and highly constructed pores and good mechanical strength capable of uplifting membrane properties and performance.

Diatoms are the most successful and abundantly available single-celled photosynthetic eukaryotic microalga found in almost all aquatic habitats across the globe [16,17]. They are very different from other microalgae in that they have a two-part siliceous cell wall, also known as frustules [18]. Their silica content varies from species to species depending upon the availability of silica and other nutrients, as well as environmental factors, such as salinity, temperature, light, and growth stage [19]. Diatom silica cell walls are highly sculpted with numerous openings (pores and slits) [20], which facilitates interaction with the external environment by the exchange of nutrients, as well soluble gases [21]. Many research groups have decoded the structural composition of the diatom cell wall and its morphological characteristics. Desikachar and Prema highlighted the diversity of diatom morphology in the Bay of Bengal and developed the Atlas of Diatoms [22]. Later, Bayer and his research group developed computer vision technology for automatic diatom identification and classification (ADIA) through digital images [23]. Mann and colleagues have characterized and identified many diatoms based on their cell wall morphology and studied their life cycles in detail [24]. Recently, a mini atlas of diatom frustules imaging revealed a stiff and strong hydrated amorphous silicate 3D nanostructured cell wall [25]. After cell death and decomposition, the diatom biosilica aggregates and forms sediment on the seafloor known as diatomaceous earth (diatomite).

Diatomite has been used as an inexpensive raw martial in various industries as an abrasive, filter material, sorbent, anticaking agent, and insulation material [26]. Diatom biosilica has been approved for various nanotechnologies [27,28] and used as semiconducting inorganic material in solar cells, batteries, and electroluminescent devices [29]. In addition, diatomite-based hybrid microfiltration carbon membranes have been used for oil decontamination in wastewater treatment [30]. Many researchers and small-scale industries are currently attracted to the mass cultivation of diatoms due to their abundance and unique porous cell wall architectures, which have a vast scope of applications in various filtration techniques.

In the present study, for the first time, we propose the use of residual biomass waste-derived natural 3D mesoporous silica from marine diatom *T. lundiana* CSIRCSMCRI 001 (accession number: MH553154) cultures as active material in polysulfone (PSF) membranes to enhance the surface functionality, to achieve unique morphology, and to impart strength and antifouling properties to the membrane. The shell (cell wall) of diatom, made of amorphous hydrated silica (frustule), consists of silicon dioxide (SiO_2_), forming an –Si–O–Si– nanostructure. The pore scaffold architecture of the frustules can hold the water molecule inside the polymer matrix, making the membrane more hydrophilic and helping to overcome membrane fouling. Moreover, the dispersed diatom frustules may create a greater number of water channels, facilitating the transport of more water through the polymer matrix. This is an important report of the hydrophilic alteration of a composite membrane using diatomite with myriad openings (pores and slits) to facilitate a water–polymer interaction. The well-constructed natural 3D mesoporous diatom frustules are expected to disperse uniformly in the PSF polymer. Diatomite was chosen as the membrane filler because of its high water absorption, large surface area, high heat, chemical resistance, and high usability [31]. The frustule of *T. lundiana* CSIRCSMCRI001 has a unique honeycomb hexagonal surface morphology with high porosity and highly constructive pore size [20]. Such a structure in the membrane may facilitate the separation of components such as antibiotics, dyes, etc. In the present study, the physicochemical characteristics of composite membranes were characterized by contact angle, ATR-IR, TGA, SEM, and AFM. The performance of the membrane was studied by measuring the pure water flux, removal of dye and antibiotics, and separation of an oil–water emulsion.

## 2. Materials and Methods

### 2.1. Diatom Cultivation and Biomass Harvest

Cultures of the diatom *Thalassiosira lundiana* CSIRCSMCRI 001 (accession number: MH553154) were grown in batch mode. In batch culture, five sterile culture flasks (5 L) were taken, and each of the flasks was filled with 2.5 L of sterilized f/2 medium [32]. About 50 mL (0.1 × 10^3^ cells/mL) of axenic unialgal diatom culture was inoculated in each flask. The inoculated flasks were kept under laboratory conditions at constant temperature (25 ± 1 °C), light intensity (100 µmol s^−1^ m^−2^), and auto-programmed light and dark period (L:D: 12:12 h) for 15 days. The cultures were kept for 2 h to settle. Then, the upper layer (transparent spent media) was recovered by siphoning without disturbing the settled biomass. Settled biomass was collected into tubes and centrifuged at 6000 rpm (Kubota-7780, INKARP Instruments Pvt. Ltd. Hyderabad, India) for 10 min to remove the remaining water. The final fresh biomass was weighed and kept at −80 °C for further study.

### 2.2. Biosilica Generation and Physicochemical Characterization

After the extraction of valuable products such as phytopigments, lipids, etc., residual biomass (spent biomass) was transferred to a 500 mL beaker, and 2 mL of hydrogen peroxide (30%) and 5 mL of concentrated nitric acids were added to the biomass and left on a hot plate at 100 °C for 1 h [33]. Acid-cleaned silica frustules were washed with tap water, followed by washing with absolute ethanol. Surface morphology and elemental composition of pure silica frustules were determined with the help of SEM-EDX analysis using JEOL JSM 7100F Thermal Field Emission electron microscope (JEOL Ltd., Akishima, Tokyo, Japan). The surface functional spectral annotations of the acid-cleaned silica frustules and vacuum-dried spent biomass (un-cleaned) residual waste were performed under the IR region according to the protocol of Trobajo and Mann [34]. X-ray diffraction analysis was carried out to determine the biosilica crystalline phases using a Philips X’pert MPD XRD system with CuKα radiation (λ = 1.54056 Å). The selected scanning speed was 2°/min. In addition, the thermal stability of diatomite was analyzed using TGA/SDTA851e (Mettler-Toledo India Private Limited, Powai Mumbai, India). gradient temperature ranging from 25 °C to 800 °C at a heating rate of 5 °C min^−1^.

### 2.3. Incorporation of Diatomite in Polysulfone (PSF): Preparation and Characterization of the Diatomite Composite Membrane

The virgin and diatomite-incorporated polysulfone composite membranes were prepared via the phase inversion method. Polysulfone (UDEL P-3500, purchased from Solvay Advanced polymers) and polyvinylpyrrolidone (Sigma-Aldrich Chemicals Private Limited, Anekal Taluk, Bangalore, India) (used as a pore former) were weighed and kept in the oven at 80 °C for 24 h. A polymer blend solution containing polysulfone (18 wt.%) and polyvinylpyrrolidone (5 wt.%) was prepared by dissolving the polymers in dimethylformamide (DMF) at 85 °C for 22 h at a stirring speed of 260 rpm. To prepare the composite membranes, the diatomite was added to the polymer blend solution in different proportions (0.05 to 0.5 wt.%). The well-prepared homogeneous PSF/PVP and PSF/PVP/diatomite solutions were kept at room temperature for 12 h to remove the air bubbles. The PSF/PVP and diatomite-modified PSF/PVP solutions were cast on non-woven polyester fabric (Filtration Science Corporation, New York, NY, USA) at a speed of 2 m per minute to achieve a thickness of 60 µm (by adjusting the gate height). The fabric coated with polymer dope solution was immersed in a water bath for precipitation. The casted membranes were washed for 72 h with deionized water to completely remove the solvent and the pore former (PVP). The membranes were stored in a freezer for further use. The membrane compositions and nomenclature are presented in Table 1.

The virgin and composite PSF membranes were characterized by surface contact angle measured by the sessile drop method. The contact angle was measured for water droplets (5 µL) with a high-megapixel color camera and analyzed using Image J freeware (version 1.48S) software. On average, the contact angle was measured for 10 droplets at different positions on a membrane sample to minimize the error. Surface and cross-sectional images of a membrane were analyzed using field emission scanning electron microscopy (FESEM, JEOL JEM 2100, JEOL Ltd., Akishima, Tokyo, Japan). Atomic for microscopy (AFM) imaging of the membranes was performed on an Ntegra Aura atomic force microscope (NT-MDT Spectrum Insruments, Moscow, Russia) in semicontact mode using an NSG 01 silicon probe. The surface roughness and 2D and 3D topography of the membranes were studied by NT-MDT surface analysis software (build 3.5.0.2064). The scan area of all the membranes was selected as 5 µm × 5 µm. The surface functionality of the membranes was analyzed by Attenuated Total Reflectance- Fourier Transform Infrared (ATR- FTIR) spectroscopy (Agilent Technologies India Pvt. Ltd., Mumbai, India) equipped with Cary 600 series FTIR microscope in the range of wavenumber 4000 to 400 cm^−1^ with 23 scans at room temperature.

### 2.4. Membrane Performance Test

The membrane’s pure water permeability (PWP) was measured by mounting the membrane circles (each with an effective area of 0.0015 m^2^) in a cross-flow filtration unit. Membrane permeability was measured at a pressure of 20 psi, and the PWP was calculated by following Equation (1).
(1)$$\mathrm{PWP}=\frac{V}{A\times t}$$
where V is the total volume of permeate in (L), A is the area of the membrane (m^2^), and t is the time (h). Pure water fluxes of the membranes were also measured at different operating pressures.

Fouling tests of the membranes were carried out at 20 psi operating pressure for a period of 7 h using a model foulant, i.e., bovine serum albumin (Sigma-Aldrich, now Merck) solution (500 mg/L). For BSA fouling experiment, the membrane flux was measured every one hour. The concentration of BSA in feed and permeate samples were determined using a UV-VIS spectrophotometer (Shimadzu UV-2700). Spectral data were recorded at 280 nm. The BSA rejection % was calculated using Equation (2).
(2)$$\%R=1-\frac{C_{p}}{C_{f}}\times 10$$
where C_p_ and C_f_ denote the BSA concentration in permeate and feed, respectively.

To determine the antibiotic rejection by the membranes, two different antibiotics, namely ampicillin (Mw. 371.39 g/mole, Sigma-Aldrich, now Merck) and neomycin (MW. 908.88 g/mole, Sigma-Aldrich, now Merck), were selected, and two different stock solutions (100 µM concentration) were prepared separately in distilled water. Antibiotic rejection tests were performed at 20 psi pressure. At a particular time, the feed and permeate samples were collected. The concentration of antibiotics in feed (AC_F_) and permeate (AC_P_) was measured with the help of UV-VIS spectroscopy at 290 nm (for ampicillin) and 320 nm (for neomycin). From the spectral analysis, the antibiotics rejection percentage (%) was calculated by using Equation (3).
(3)$$R\%=\frac{{\mathrm{AC}}_{F}-{\mathrm{AC}}_{p}}{{\mathrm{AC}}_{F}}\times 100$$
where AC_F_ and AC_P_ are the concentrations of antibiotics (ampicillin/neomycin) in feed and permeate, respectively.

Motor crude oil was selected to study the separation of an oil–water mixture using the developed membranes. The motor oil–water emulsion was prepared by adding 1000 ppm motor crude oil to 1 L of distilled water and stirring at 4000 rpm for 24 h in sodium lauryl sulfate (SLS) surfactant. The oil rejection tests were performed at 20 psi pressure. Both feed and permeate samples were collected. The concentration of oil in feed (C_F_) and permeate (C_p_) was measured with the help of UV-VIS spectroscopy at a wavelength of 270 nm [34]. From the spectral data, the % of oil rejection was calculated using Equation (4).
(4)$$R\%=\frac{{\mathrm{OC}}_{F}-{\mathrm{OC}}_{p}}{{\mathrm{OC}}_{F}}\times 100$$
where OC_F_ and OC_P_ are the concentrations of oil in feed and permeate, respectively.

To determine the dye rejection performance of diatomite composite membranes, 10 mL of 0.1 M rhodamine 6 G (MW. 479.02 g/mol) stock solution was prepared in distilled water. From this stock solution, 1 L of 15 µM solution was prepared, and Rho 6G rejection tests were performed at 20 psi pressure. At a particular time, both feed and permeate samples were collected. The concentration of Rho 6G in feed (RC_F_) and permeate (RC_p_) was measured with the help of UV-VIS spectroscopy at a wavelength of 530 nm. From the spectral data, the Rho 6G rejection % was calculated using Equation (5).
(5)$$R\%=\frac{{\mathrm{RC}}_{F}-{\mathrm{RC}}_{p}}{{\mathrm{RC}}_{F}}\times 100$$
where RC_F_ and RC_P_ are the concentrations of dye (rhodamine 6G) in the feed and permeate, respectively.

## 3. Results and Discussion

### 3.1. Cell Growth, and Production of Biomass, Lipids Productions

The density of cultures (cell number) increased exponentially and attained maximum growth (1.75 × 105 cells/mL) within 10 days in the batch cultures. After that, cell numbers were saturated for up to 15 days with a small fluctuation system (Figure 1) representing the depletion of essential nutrients, i.e., P, N, and trace metals levels in the media. Many researchers have shown that the content of nitrate/nitrite (N) [35], phosphorous (P) [36], and trace metals [37] were significantly decreased with increased cell number and incubation time in batch cultures. Nutrient levels are primary factors indirectly proportional to the cell growth in the batch cultures [38]. Live cultures of the marine diatom *T. lundiana* CSIRCSMCRI 001 were brown under a light microscope (Figure 2a), which corresponds to the presence of a high percentage of lipids, as well as fucoxanthin. In total, 10 g of fresh biomass was produced from 12 L culture within a 15-day cultivation period. From this, 2.5 g (25%) of total lipid was extracted, followed by 1.25 g (12.5%) of natural 3D porous biosilica or diatomaceous earth (Diatomite/kieselguhr) produced in acid cleaning.

### 3.2. Morphology and Physiological Features of Diatomaceous Earth

Morphologically, the diatom *T. lundiana* isolate CSIRCSMCRI 001 shows 3-dimensional futures and belongs to the centric type. Under light microscopy, individual cells appeared in a square. This is maybe due to the 3D effect in liquid media (Figure 2a). However, the higher magnification (100×) and scanning electron micrographic examination confirmed that the siliceous frustules were slightly convex and special in shape. The frustules size ranged from 17 to 25 µm, and the per-valve axis was 5 to 10 µm. Spine-like structures called rimoportula were scattered on the surface of the valves (Figure 2b,c). Structurally, the surface of the siliceous frustule appears in hexagonal sculpts with 8 to 10 uniform pore size of 20 to 22 nm (Figure 2d). These correspond to the previous results reported by Fryxell, who reported the diatom *T. lundiana* from the north-central Gulf of Mexico and assumed that it was distributed in temperate to subtropical waters [39].

From infrared spectral annotation, the surface functional group stretching of vacuum-dried biomass shows peaks at 550 cm^−1^ for Al–OH bands, 1420 cm^−1^ for –C–O stretching in carbohydrates, 1600 cm^−1^ for >C=O stretching of ester or fatty acids, and 2900 cm^−1^ for –CH_2_– banding. These stretches are disappeared during acid cleaning of silica frustules. Similarly, a wide peak range was found from 3250 cm^−1^ to 3750 cm^−1^ specifics for free –OH and N-H groups, indicating that the surface of the diatom is covered with –OH free functional biomolecules, such as polysaccharides proteins. In the case of acid-cleaned biomass residues, Al–OH, –C–O, >C=O, –CH_2_–, –OH, and N–H band stretch intensities were decreased, confirming that the diatoms that covered polysaccharides, proteins, or phospholipids were degraded during acid treatment (Figure 3a). However, a stretch at 950 cm^−1^ for Si–OH bonding is only seen in acid-cleaned biomass residues. A strong stretch was seen at 480 cm^−1^ for Si-O banding and at 1100 cm^−1^ for Si–O–Si banding in the case of both vacuum-dried spent biomass and acid-treated silica frustules. This confirmed that the diatom cell wall comprises a high percentage of silicate (SiO_2_). X-ray diffraction also confirmed that the crystalline diatomite powder is composed of silicon dioxide (SiO_2_) (Figure 3b).

### 3.3. Thermal Gravimetric Analysis (TGA) and ATR-IR Spectra of the Composite Membranes

The thermal stability of the membranes increased with increased diatomite percentage (Figure 4a). According to TG analysis, the pure diatomite material showed only 8% mass reduction at 100 °C, followed by 42% mass reduction at 400 °C. Similarly, the control PSF membrane (without diatomite) had a thermal stability of up to 480 °C, with a mass reduction is ~10%. Additionally, a 32% mass reduction was found for the PSF membrane at 500 °C. The diatomite-embedded composite membranes had better thermal stability than control membranes (Figure 4a). In particular, the PSF/Dia_0.5_ membrane showed the highest thermal stability, with negligible weight loss up to 500 °C. Thereafter, only ~25% of mass reduction was noticed at 800 °C. From these results, it can be concluded that the thermal stability of PSF composite membranes increased with increased percentage of diatomite. These results are in agreement with the observations noted in previous reports [30] that included commercially available diatomite material with 300–500 nm pore size and 20 to 40 µm particle diameter. According to these reports, the microfiltration carbon membrane’s thermal stability was increased by increasing the percentage of diatomite from 0 to 15%. In the present study, for the acid-cleaned silica frustule, the pore size varies from 20 to 200 nm, and particle diameter is in the range of 15 to 20 µm (Figure 2). The addition of silica frustules with these structural features supports comparatively less mass reduction, even at high temperatures, leading to the higher thermal stability of PSF/Dia composite membranes.

The ATR-IR spectrum of the membranes is shown in Figure 4b. The peak at 1151 cm^−1^ is the benchmark for sulfone stretching (O–S–O) in the polysulfone membranes. Similarly, peaks at 1244 cm^−1^ and 1585 cm^−1^ are attributed to C–O–C and aromatic –C–C– bonding. In addition, the bands of the polysulfone aromatic ring are recorded at 1020 cm^−1^ and 830 cm^−1^ for C–H stretching. It is not easy to find distinct peaks of diatoms, as they have been incorporated in minimal amounts in the membranes. However, for diatom-embedded composite membranes (PSF/Dia_0.1_, PSF/Dia_0.2_, and PSF/Dia_0.5_), the characteristic peak for –OH stretching was increased at 3750 cm^−1^, which is absent in virgin PSF/Dia_0.0_ membranes (Figure 4b). In addition, the ATR-IR peak at 952 cm^−1^ was attributed to Si–OH stretching for diatomite mesoporous silica material in the composite membranes. ATR-IR spectral results confirmed that the effective formation of –OH groups was increased, which can promote the surface hydrophilicity of the composite membrane.

### 3.4. Membrane Hydrophilicity

The water contact angle is a direct measure of the membrane surface hydrophilicity, which influences any membrane’s flux and antifouling properties. The wettability of the surface of the composite membrane is expected to improve with the presence of hydroxyl groups of diatomite. It is possible that while preparing the membrane by phase inversion processing (exchange of solvent with non-solvent), diatomite may migrate towards the membrane surface because of its great affinity towards water. The contact angle values of different membranes are given in Table 2. It is observed from the table that the contact angles of the membranes decreased in the order PSF/Dia_0.0_ > PSF/Dia_0.05_ > PSF/Dia_0.1_ > PSF/Dia_0.2_ > PSF/Dia_0.5_. This indicates that the control membrane (without diatomite) exhibits the least hydrophilicity (contact angle, 86.1°), whereas the composite membrane with the highest dose of diatomite (0.5%) shows the highest hydrophilicity (contact angle, 59.5°). The contact angles of the other composite membranes lie between these two extremes. A pictorial presentation of the shape of water droplets on different membrane surfaces is shown in Figure 5. The membrane wettability is dependent on the contact angle. The contact angle and surface wettability are inversely proportional (Figure 5). In the presence of diatomite material, the wettability of the membrane (hydrophilicity) was increased, indicating that the incorporation of diatomite may enhance membrane hydrophilicity.

### 3.5. Membrane Morphology

#### 3.5.1. Surface Roughness and Topography by Atomic Force Microscopy (AFM)

AFM was used to study both the 2D and 3D surface morphology of the control and composite membranes; AFM images are presented in Figure 6. Surface roughness plays a pivotal role in controlling membrane permeability and fouling properties. The investigated membranes manifest nodular morphology with regular hills and valleys. The white dots represent hills, whereas the black dots represent valleys. The average and root mean square roughness were measured, and the values are presented in Table 2. The membranes roughness values of diatomite-embedded composite showed a decreased order of 4.81 nm, 4.79 nm, 4.19 nm, and 3.62 nm for PSF/Dia_0.05_, PSF/Dia_0.1_, PSF/Dia_0.2_, and PSF/Dia_0.5_, respectively. However, for control membranes the value was the highest (5.73 nm) as compared to others. The fact that diatomite-embedded composite membranes are characterized by lower roughness could alleviate adhesion of the foulant on the membrane surface and thus reduce fouling.

#### 3.5.2. Surface and Cross-Sectional Morphology by Scanning Electron Microscopy (SEM)

Surface (50,000×) and cross-sectional (500×) morphology of the membranes were studied by SEM, and the images are displayed in Figure 7 and Figure 8, respectively. The surfaces of all the membranes are porous, and interestingly, the pore size was increased with an increased % of diatomite loading in the membranes. The pore size of the membranes was measured by using the Image J software tool, and it was observed that the pore size of the control PSF/Dia_0.0_ membrane was 39 ± 10 nm, whereas for PSF/Dia_0.1_, PSF/Dia_0.2_ and PSF/Dia_0.5_ membranes, the pore size values were 59 ± 6 nm, 76 ± 5 nm, and 106 ± 5 nm, respectively. As diatomite has an excellent water affinity, it is possible that during the phase inversion process, it attracts more water, and as a result, phase inversion is faster and abrupt, resulting in bigger pores. Hence, diatomite plays a significant role in controlling the pore morphology of the membranes.

The cross-sectional morphology of the membranes was studied at higher resolution using SEM, and the images are presented in Figure 8. According to cross-sectional image views, all the membranes have a typical asymmetric structure with a dense top layer, supported on a comparatively more porous sublayer in the middle, with the bottom-most layer containing large macro voids. The control PSF/Dia_0.0_ membrane displays circular pore morphology (Figure 8a,(a1)), whereas the diatomite-embedded membrane exhibits elongated pore morphology and macro voids (Figure 8b–d,(b1–d1)). With the increase in the quantity of diatomite, more and more macro voids are observed. Cross-sectional morphology also reveals the presence of silica frustules in the interstices of the membranes (Figure 8(b1,c1,d1)). The number of frustules per unit area increases with increased concentration of diatomite in membranes. The addition of diatomite to the PSF membranes also results in higher pore volume (Figure 8c,(c1)).

The PSF/Dia_0.5_ membranes exhibit the loosest pore morphology with the largest pores and most macro voids among all the membranes studied here. This is because diatomite is hydrophilic; hence, its presence in the polymer matrix attracts water, leading to the rapid exchange of solvent (DMF) and non-solvent (water) during the immersion precipitation process [40]. SEM energy-dispersive X-Ray (SEM-EDX) analysis was used to determine the absence of sulfur (S) on the surface of silica frustules before embedding in PSF membranes (Figure 9). Frustules embedded in PSF membranes have S on their surface. These results confirmed the interactive banding between silica (SiO_2_) of frustules and free sulfur groups of PSF/PVP polymers (Figure 10). The precipitation rate increased with the increase in the concentration of diatomite, which is conducive to porous structures.

Energy-dispersive X-Ray (EDX) spectroscopic analysis of the membranes was carried out (Figure 10a). The surface elemental mapping of silica frustules denotes their elemental composition, and concentrations are shown in Figure 10b. The images in blue, green, and red were taken using SEM-EDX, exhibiting the elemental distribution of carbon, sulfur, and silica, respectively, on the surface of silica frustules. Cross-sectional morphology of the PSF/Dia membranes shows silica frustules embedded in the polymer matrix.

### 3.6. Membrane Performance

#### Pure Water Flux and Protein Rejection

The two most important properties of any ultrafiltration membrane are high flux and good antifouling properties. The control membrane, i.e., PSF/Dia_0.0_, has the lowest flux (224.5 LMH at 20 psi operating pressure), whereas at the same operating pressure, the diatomite composite membranes possess better flux than control (Figure 11a). The fluxes are 320.8 LMH, 432.4 LMH, 558.9 LMH, and 806.8 LMH for PSF/Dia_0.05_, PSF/Dia_0.1_, PSF/Dia_0.2_, and PSF/Dia_0.5_, respectively. Hence, incorporating diatomite enhances the membrane flux from 224.5 LMH (Control) to 806.8 LMH (PSF/Dia_0.5_ membrane). The presence of porous inorganic silica frustules in the PSF membrane increases the number of pores and pore patterns (Figure 7 and Figure 8) and also enhances membrane hydrophilicity [41]. Because of these dual effects, flux of composite membrane increases.

Moreover, incorporating diatomite in the PSF matrix results in the distribution of silica frustules in the polymer matrix, and these silica frustules enhance the distance between polymer chains. Such phenomena generate membrane pores and voids, leading to better water permeability of the composite membranes [14]. The enhancement of flux is in agreement with the hydrophilicity and surface roughness of the membranes, as higher hydrophilicity and lower roughness always boost membrane flux (Table 2).

It is known that membrane flux is directly proportional to the applied transmembrane pressure, i.e., increasing the operating pressure increases the driving force for water permeation through the membrane. The pure water flux of the control PSF/Dia_0.0_ membrane is 125 LMH at 5 psi operating pressure (OP), whereas at 30 psi OP, the water flux is increased to 355 LMH. Similar results are observed for diatomite composite membranes. The PSF/Dia_0.5_ membrane flux was 545.6 LMH at 5 psi OP, whereas at 30 psi OP, the water flux was increased to 1268 LMH (Figure 11b)**.** Higher operating pressure enhances the membrane flux; hence, it reduces the concentration polarization and mitigates the common interaction of the solutes with the membrane surface. As a result, membrane fouling decreases.

A membrane fouling study was carried out with a BSA protein solution. The BSA solution (500 mg/L) was passed through the membranes continuously for seven hours (7 h) at 20 psi operating pressure. BSA rejection was measured every hour by analyzing the concentration of BSA in the permeate solution (Figure 11c). The BSA rejection was 66 ± 0.50% for the control (PSF/Dia_0.0_) membrane, whereas the rejection values were 70 ± 1.0%, 80 ± 0.50%, 94 ± 0.50%, and 98.5 ± 0.075% for PSF/Dia_0.05_, PSF/Dia_0.1_, PSF/Dia_0.2_, and PSFDia_0.5_, respectively. As the composite membranes contain polysulfone with silica frustules, the sulfonyl (–SO_2_–) functionality of polysulfone and the negative functionality of silica frustules (SiO_2_) impart a negative charge on the membrane and make the membrane more hydrophilic. Such a hydrophilically charged surface prohibits BSA molecules from being adsorbed on the membrane surface. A trivial decrease in flux with time is observed for all the composite and control membranes. However, the diatomite composite membranes have better flux than the control (PSF/Dia_0.0_) (Figure 11c).

The rejection profile of crude motor oil by the control and composite membranes is shown in Figure 11d. The control membrane exhibited only 20% oil rejection, whereas the composite membranes exhibited much higher oil rejection than the control. Moreover, the percentage of oil rejection increased with the percentage of diatomite in the composite membrane. For the diatomite composite membranes (DUFs), oil rejection increased from 49% to 81%, increasing the diatomite from 0.05 to 0.5% (Figure 11d). SLS facilitated oil removal by PSF/Dia_0.5_ membrane, showing the best oil rejection among the five tested membranes, which may be due to its high hydrophilicity. Because this membrane was more hydrophilic than the PSF/Dia_0.0_ membrane, it restrains oil from passing through it. In contrast, the control membrane, i.e., polysulfone, is hydrophobic; hence, its oil retention rate was always lower than that of the other composite membranes.

Antibiotic, i.e., ampicillin and neomycin, rejection of the membranes is shown in Figure 12a,b. The diatomite composite membranes showed better antibiotic retention than the control membrane (PSF/Dia_0.0_). The rejection percentages of antibiotics (ampicillin and neomycin) were increased with an increased percentage of diatomite in the membrane. The antibiotic rejection efficiency ranked in the following order: PSF/Dia_0.0_ < PSF/Dia_0.05_ < PSF/Dia_0.1_ < PSF/Dia_0.2_ < PSF/Dia_0.5_ (Figure 12). The rejection percent of ampicillin for PSF/Dia_0.05_, PSF/Dia_0.1_, PSF/Dia_0.2_, and PSF/Dia_0.5_ membranes was 18.36%, 32.22%, 58.25%, and 75.84% respectively. For neomycin, the values were 36.28%, 45.25%, 66.96%, and 85.88%, respectively. For the control membrane, i.e., PSF/Dia_0.0_, rejection values of ampicillin and neomycin were 8.16% and 26.29%, respectively. The higher rejection of these antibiotics found in case of diatomite composite membranes might be due to surface charge of the membranes. Diatom-based silica material has negative zeta potential values [42]. Hence, incorporation of diatomite material in the PSF membrane is expected to impart a negative potential on the membrane. Higher content of diatomite might induce higher negative membrane potential. Structurally, ampicillin has a carboxylic acid group, and neomycin has a dihydroxy functional group. Hence, Donnan exclusion prevails between the membrane surface and ampicillin/neomycin. Because of this, the mutual repulsion membrane rejects these antibiotics. The rejection of antibiotics of the composite membranes was higher than that of the control membrane, which was devoid of diatomite.

Among the composite membranes, PSF/Dia_0.5_ has the highest content of diatomite. Hence, it is expected to possess the highest negative potential, and that is why it shows the highest rejection among the composite membranes. It is also observed that the rejection of neomycin (MW. 614.64 g/mole) is greater than that of ampicillin (MW. 349.39 g/mole). This is because of the higher molar mass of the former relative to the latter. This indicates that in addition to the Donnan exclusion, the sieving mechanism also contributes to antibiotic rejection.

Rhodamine 6G was selected to study the dye rejection of the membrane; dye rejection values of the membranes are shown in Table 3. Rho 6G rejection follows the order of PSF/Dia_0.0_ < PSF/Dia_0.05_ < PSF/Dia_0.1_ < PSF/Dia_0.2_ < PSF/Dia_0.5_. This indicates that the control membrane, devoid of diatomite, shows the lowest rejection of the dye (45.19 ± 1.02%). The incorporation of diatomite in the membrane enhanced dye rejection from 51.30 ± 1.51% to 94.84 ± 1.52%. The same results were observed by Koyuncu et al., who reported that the adsorption of Rhodamine B onto the natural diatomite is a spontaneous, endothermic process [43]. Both Rhodamine B and Rhodamine 6G dyes are derivatives of xanthene; it is expected that Rhodamine B and 6G will be adsorbed onto the diatomite. Current experiment study results showing that the diatomite adsorbs the pollutants might be due to endothermic phenomena. We observed that as the diatomite content increased in the membranes, the dye rejection was increased.

## 4. Conclusions

The diatom *T. lundiana* CSIRCSMCRI 001, 25% lipid, and 12.5% 3D mesoporous biosilica (diatomaceous earth/diatomite) were produced by cultivation and biomass harvesting method. Acid-cleaned 3D mesoporous biosilicate was successfully used in PSF ultrafiltration membranes with the phase inversion method. The performance of the diatomite composite membrane was examined, together with the nascent polysulfone membrane. The PSF/diatomite composite membranes exhibit excellent hydrophilicity, large pore voids, and low surface roughness through the interaction of silicon dioxide with the polysulfone matrix. The diatomite composite membranes possess a better flux of 806.8 LMH at a lower pressure of 20 psi compared to the control PSF membranes. The order of rejection efficiency of antibiotics, i.e., ampicillin and neomycin, is PSF/Dia_0.0_ < PSF/Dia_0.05_ < PSF/Dia_0.1_ < PSF/Dia_0.2_ < PSF/Dia_0.5_. Higher loading of the PSF/Dia0.5 membrane enhanced water holding capacity (42.53), BSA rejection (98.52), and removal of rhodamine 6G (94.84). The prepared composite membranes may have ample scope to treat effluents for the removal of oil, dye, and pharmaceutical waste from water bodies.

## Figures and Tables

**Figure 1 polymers-14-01750-f001:**
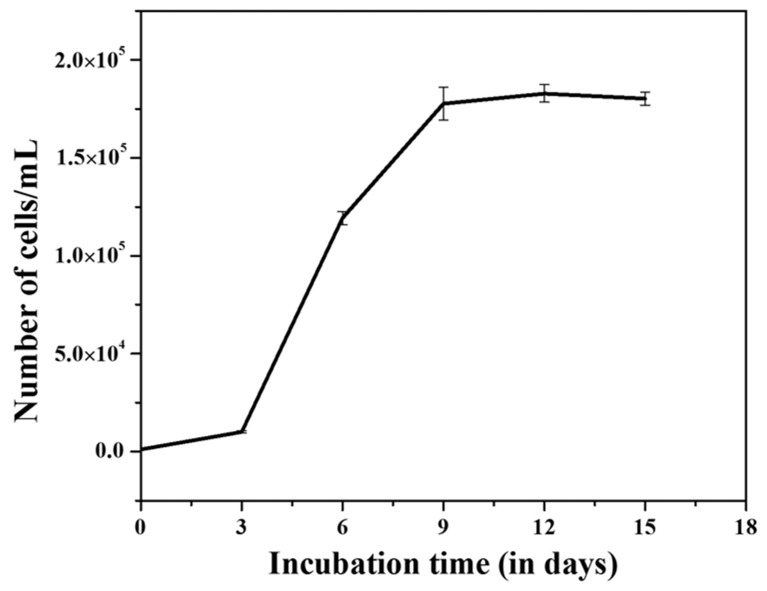
Growth response of *T. lundiana* CSIRCSMCRI 001 in modified f/2 media under laboratory conditions.

**Figure 2 polymers-14-01750-f002:**
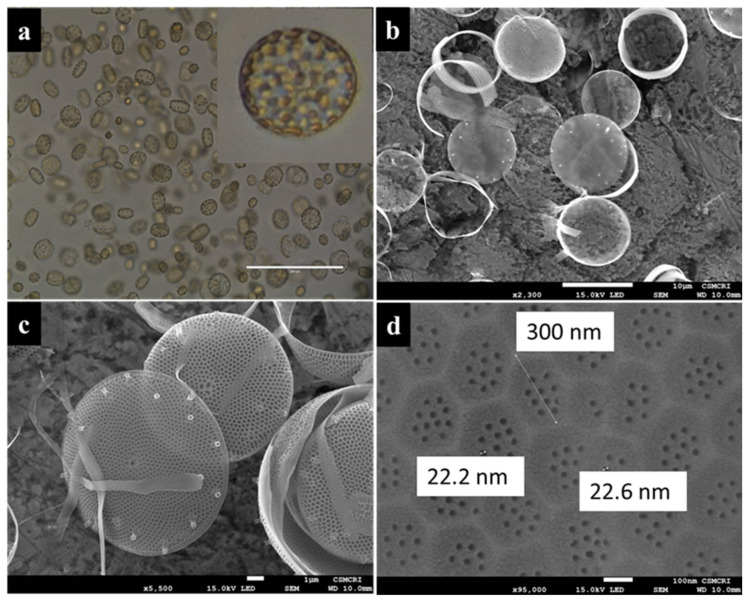
Microscopic images of diatom *T. lundiana* CSIRCSMCRI 001 (**a**). Unialgal culture live photos under light microscopy at 40× and 100×; (**b**,**c**) acid cleaned diatoms cells; (**d**) silica frustules surface appears as hexagonal sculpts with 8–10 pores in 20–22 nm diameter (scale 100 nm).

**Figure 3 polymers-14-01750-f003:**
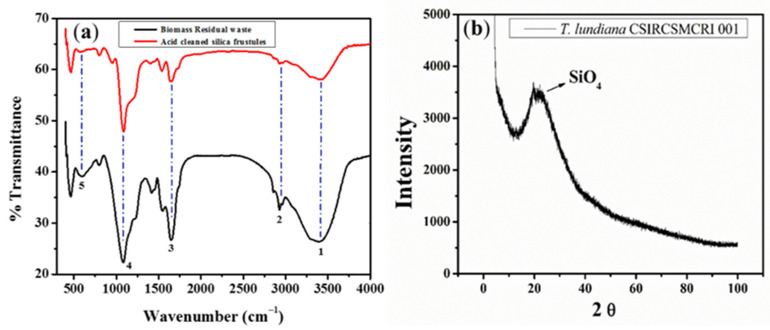
Spectral annotation of the diatom *T. lundiana* CSIRCSMCRI 001. (**a**) FTIR spectral response of vacuum-dried spent biomass waste (black colure spectra) and acid-treated biomass residues (red line spectra). (**b**) XRD pattern of purified diatomite powder (Keiselgur).

**Figure 4 polymers-14-01750-f004:**
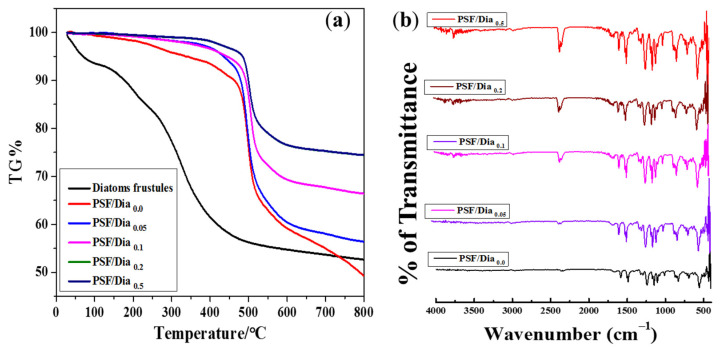
Physical characteristics of membranes. (**a**) Thermal stability and (**b**) ATR-FTIR spectra of control and diatomite-embedded PSF composite membranes.

**Figure 5 polymers-14-01750-f005:**
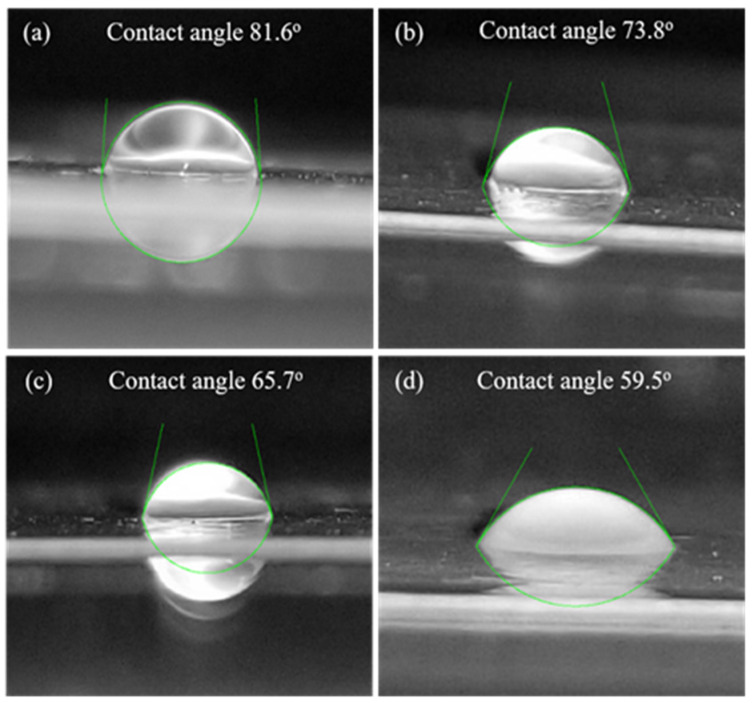
Image of water droplets on different membrane surfaces and their contact angle values. (**a**) Control membrane (PSF/Dia_0.0_) and composite membranes with different % of diatomite material: (**b**) PSF/Dia_0.1_, (**c**) PSF/Dia_0.2_, and (**d**) PSF/Dia_0.5_.

**Figure 6 polymers-14-01750-f006:**
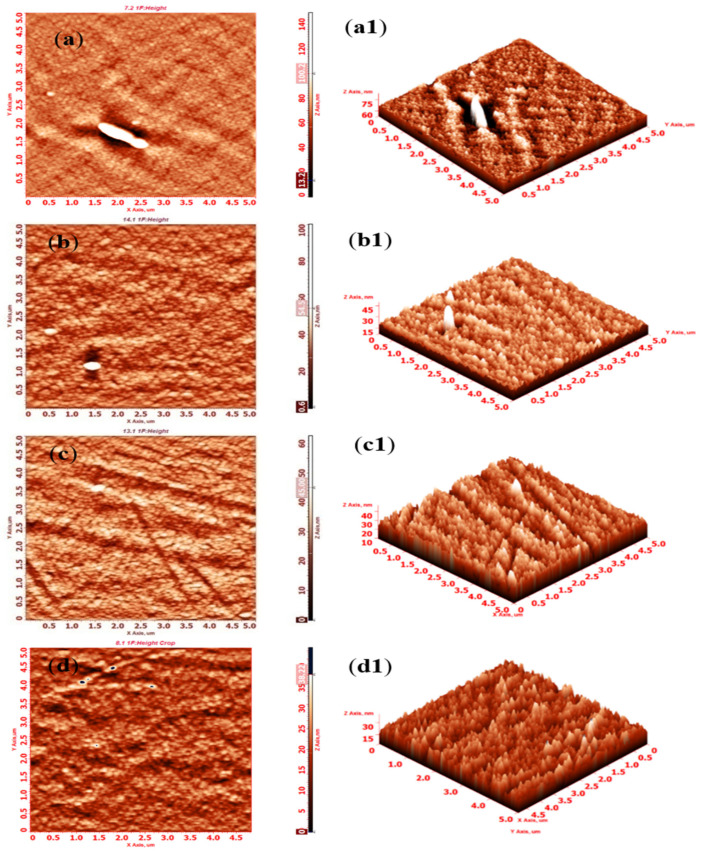
2D (**a**–**d**) and 3D (**a1**–**d1**) surface images of control (PSF/Dia_0.0_) and diatomite-incorporated PSF/Di composite membranes by AFM. (**a**,**a1**) PSF/Dia_0.0_, (**b**,**b1**) PSF/Dia_0.1_, (**c**,**c1**) PSF/Dia_0.2_, and (**d**,**d1**) PSF/Dia_0.5_.

**Figure 7 polymers-14-01750-f007:**
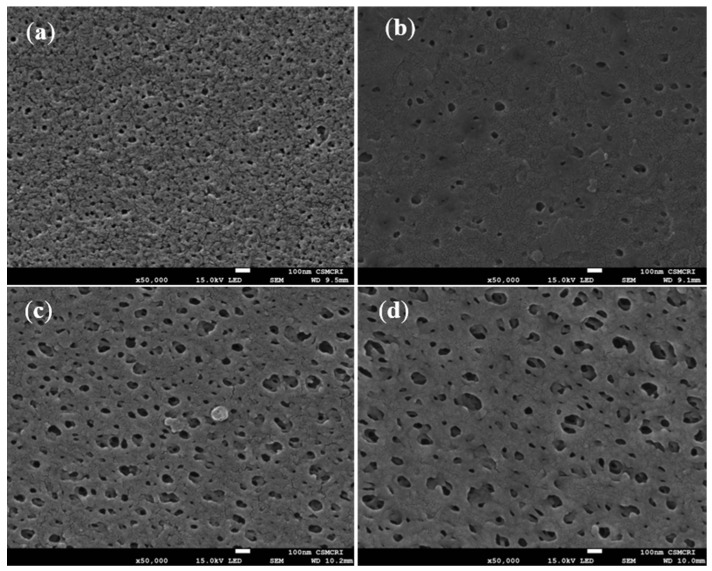
Scanning electron microscopy (SEM) images of the surface porosity of membranes at 50,000× magnification: (**a**) PSF/Dia_0.0_, (**b**) PSF/Dia_0.1_, (**c**) PSF/Dia_0.2_, and (**d**) PSF/Dia_0.5_.

**Figure 8 polymers-14-01750-f008:**
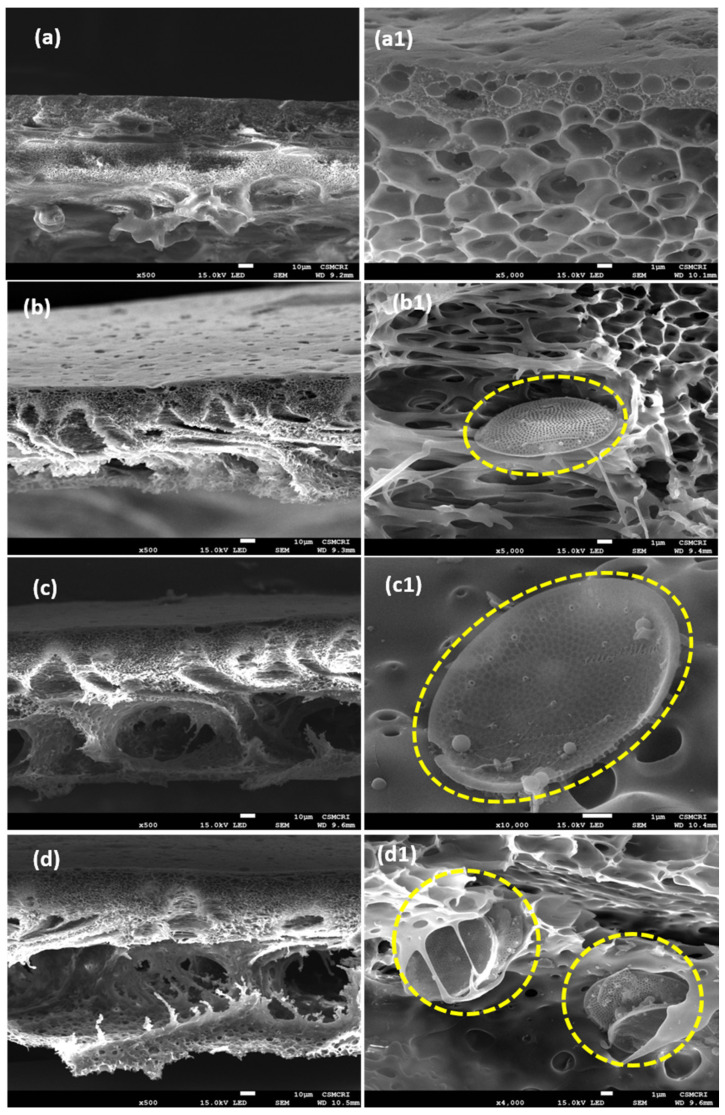
Cross-sectional morphology of diatomite hybrid membrane. SEM images on the left side represent the control, (**a**) PSF/PVP/Dia_0.0_, and diatomite hybrid membranes, (**b**) PSF/Dia_0.1_, (**c**) PSF/Dia_0.2_, and (**d**) PSF/Dia_0.5_, at ×500 magnification. Images on the right side (**a1**–**d1**) were scanned at higher magnification (×5000), denoting the presence of diatom silica frustules in membrane cross sections of (**b**–**d**), except in the control (**a1**).

**Figure 9 polymers-14-01750-f009:**
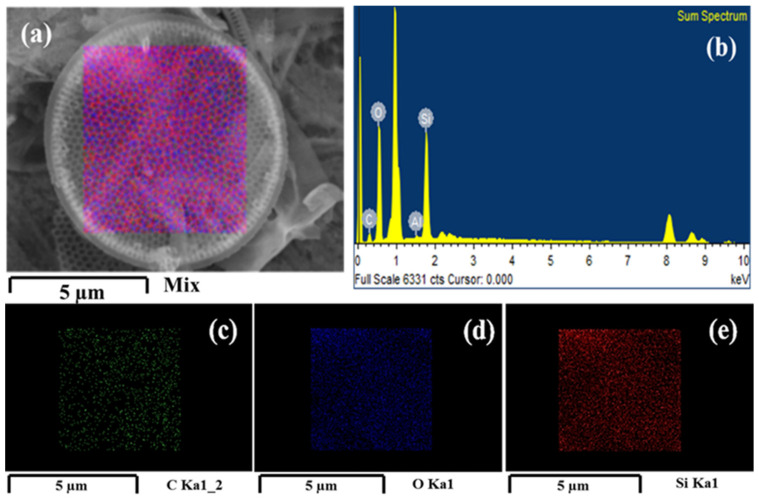
SEM-EDS imaging and surface elemental mapping of diatomite frustules. (**a**) Mapping of silica frustules; (**b**); EDS spectrum; (**c**–**e**) elemental distribution mapping for carbon, oxygen, and silicon on silica frustules, respectively.

**Figure 10 polymers-14-01750-f010:**
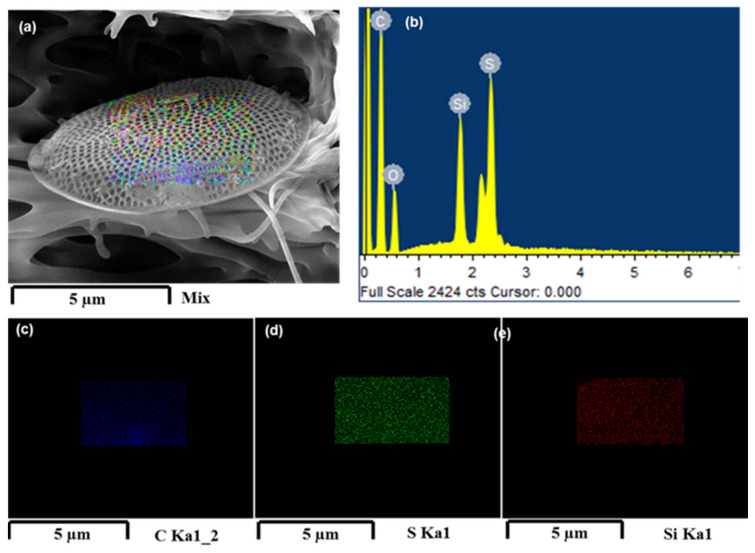
SEM-EDX imaging and mapping of silica frustules embedded in PSF/Dia_0.5_ membrane. (**a**) Cross-sectional image, (**b**) EDX spectrum, and (**c**–**e**) elemental mapping on silica frustules: (**c**) carbon, (**d**) sulfur, and (**e**) silicon.

**Figure 11 polymers-14-01750-f011:**
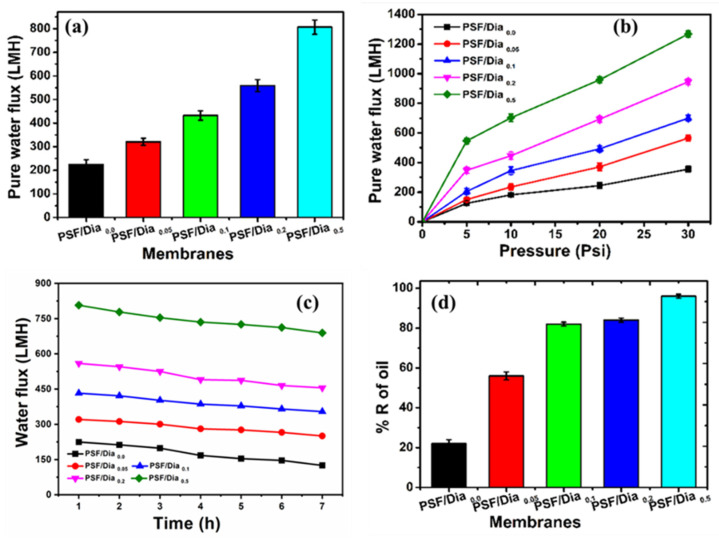
Performance of the membranes: (**a**) pure water flux, (**b**) pure water flux at different operating pressures, (**c**) fouling study, and (**d**) motor oil rejection.

**Figure 12 polymers-14-01750-f012:**
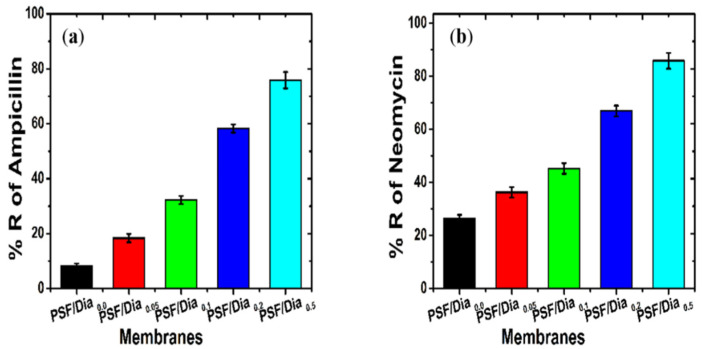
Biomedical pollutants rejection of diatomite composite membranes as compared to control (**a**) Ampicillin and (**b**) Neomycin rejection potential.

**Table 1 polymers-14-01750-t001:** Polymer dope solution composition and nomenclature of membranes.

Composite Membrane	PSF wt.%	PVP wt.%	Diatomite wt.%	DMF wt.%
PSF/Dia_0.0_	18	5	0.0	77.02
PSF/Dia_0.05_	18	5	0.05	76.95
PSF/Dia_0.1_	18	5	0.1	76.94
PSF/Dia_0.2_	18	5	0.2	76.81
PSF/Dia_0.5_	18	5	0.5	76.53

Note: DMF—dimethylformamide, PSF—polysulfone, PVP—polyvinylpyrrolidone, PSF/Dio—polysulfone and diatomite composites, wt.%—weight in percentage.

**Table 2 polymers-14-01750-t002:** Contact angle and surface roughness values of diatomite-embedded PSF composite membranes.

Membrane	Contact Angle (°)	Roughness (nm)	
		Average Roughness (S_a_)	Root Mean Square Roughness (RMS) (S_q_)
PSF/Dia_0.0_	86.1 ± 1.51	5.73	9.60
PSF/Dia_0.05_	78.5 ± 1.53	4.81	6.44
PSF/Dia_0.1_	73.8 ± 0.50	4.79	6.04
PSF/Dia_0.2_	65.7 ± 0.54	4.19	5.30
PSF/Dia_0.5_	59.5 ± 0.51	3.62	4.32

**Table 3 polymers-14-01750-t003:** Water holding capacity, BSA, and rhodamine 6G rejection by the membranes.

Membrane	Water Holding (%)	% R of BSA	% R of Rhodamine 6G
PSF/Dia_0.0_	33.50 ± 0.50	66.02 ± 0.50	45.19 ± 1.02
PSF/Dia_0.05_	35.52 ± 0.05	70.06 ± 1.01	51.30 ± 1.51
PSF/Dia_0.1_	36.51 ± 0.05	80.04 ± 0.50	60.40 ± 1.50
PSF/Dia_0.2_	37.59 ± 0.07	94.03 ± 0.52	70.61 ± 1.54
PSF/Dia_0.5_	42.53 ± 0.05	98.52 ± 0.07	94.84 ± 1.52

BSA—Bovine Serum Albumin, % R–the percentage of rejection.

## Data Availability

The data presented in this study are available upon request to the corresponding author.

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
