# Peer review of "3D Natural Mesoporous Biosilica-Embedded Polysulfone Made Ultrafiltration Membranes for Application in Separation Technology"

_polymers, 2022, doi:10.3390/polym14091750_

Round 1

Reviewer 1 Report

Please find the advice in the attached file

Author Response

Reviewer 1

  1. In the present format of the manuscript, a schematic diagram of the molecular structure is necessary to show the chemical position and three-dimensional molecular structure, which can make readers understand more about the biosilica material.

Ans).  Thanks for your kind suggestion, the Graphical abstract is modified and 3D molecular structures are used for clear visibility.

  1. The text needs to be modified, for example, the words ‘SEM-EDX, TGA, and ATR-IR’ are abbreviations, and the original statements of the words need to be indicated when they first appear.

Ans). In the revised manuscript all the abbreviations are expanded original statements of the words where they first appeared. Thank you. 

  1. In Table 3, the data show that as the content of the filler was increased, the water contact
    angle was decreased, and the surface roughness also declined. However, it is known that the roughness can intensify the hydrophilicity if the surface is hydrophilic. Therefore, combining the water contact angle and the surface roughness, it is hard to understand the evolution of the hydrophilicity of the membrane surface.

Ans). The silica frustules are hydrophilic in nature, and the addition of fillers can enhance the hydrophilicity of the membrane. In fact, increasing hydrophilicity has a much greater negative effect on hydrophilicity than decreasing roughness.

  1. How about the mechanical strength of the membranes? It can be seen from Fig. 8, that the membrane cross-section structure was gradually damaged by adding the fillers.

Ans). We are thankful to the reviewer for raising such a valid point. In general incorporation of filler reinforces a polymer matrix up to certain loading. In this case, we are generating a porous polysulfone matrix with embedded filler particles through the phase inversion method. It is observed from the morphological studies that such incorporation results in the enhancement of membrane voids. Such voids facilitate the membrane flux but extensive voids make the membrane weaker from viewpoint of mechanical strength.  The present study has been carried out with different filler loadings and it has been found that at a filler loading of 0.5 wt.% the membrane exhibits an optimum balance of permeability with its mechanical strength. The permeation test of these membranes at 20 psi on a continuous mode for a period of 7 days exhibits a steady flux without membrane failure indicating adequate mechanical strength of the membrane. However, an increase in filler loading beyond -0.5 wt.% results in membrane failure due to excessive voids.

Reviewer 2 Report

The paper "Natural 3D mesoporous bio-silica embedded Polysulfone ultrafiltration membranes for separation applications" introduces an interesting study on  diatomite composite membranes based on micro-composite inorganic bio-silica / PSf that could be used for various filtration applications - for BSA, antibiotics and a oil-water mixture (motor crude oil / water).

The paper is well presented and designed. However, there are few minor issues that should be addressed.

  1. Use in a consistent manner the notation L for liter and mL for milliliter.
  2.  Line 1213 put correctly the superscript for C deg, as 0C.
  3. Complete the provenience for BSA, antibiotics, rhodamine. 
  4. Line 291, 295, 296 use correctly the superscript for cm-1
  5. The allegation in line 295 regarding the presence of -OH and -NH groups is forced somehow. Maybe it would be helpful to put in a increased "window" the area for the specified wavelengths.
  6. Table 2 - last column is missing the data.
  7. Line 332 - please complete correctly the Table 2.
  8. Table 3 - Please use the same precision for the parameters value and the associated errors, i.e. 33,50+/- 0.50; or 70.61+/- 1.50.
  9. Line 457 - when expressing the MW values use the same precision throughout all manuscript - decide if using 2 or 3 decimals.
  10.  The Conclusions section has to be improved.

Author Response

Reviewer 2

  1. Use in a consistent manner the notation L for liter and mL for milliliter.

Ans). Thanks for the great suggestions, the manuscript was thoroughly revised in track mode and all unit notations were corrected.

  1. Line 1213 put correctly the superscript for C deg, as 0C.

Ans). In the revised MS all units’ notations are corrected in track mode.

  1. Complete the provenience for BSA, antibiotics, and rhodamine.

Ans). The provenience of BSA, and antibiotics are mentioned in material methods.

  1. Line 291, 295, 296 use correctly the superscript for cm-1.

Ans). In the revised MS all units’ notations are corrected in track mode, Thank you.

  1. The allegation in line 295 regarding the presence of -OH and -NH groups is forced somehow. Maybe it would be helpful to put in an increased "window" the area for the specified wavelengths.

Ans). Thank you for your keen observation however the -OH stretching was observed at 3750 cm-1 in presence of diatomite materials in the membranes. While -NH stretching was absent.

  1. Table 2 - the last column is missing the data.

Ans). The missing data is updated in revised MS, Thanks for your keen observations.

  1. Line 332 - please complete correctly the Table 2.

Ans). Yes, this data is updated in Table 2. Thank you.

  1. Table 3 - Please use the same precision for the parameters value and the associated errors, i.e. 33,50+/- 0.50; or 70.61+/- 1.50.

Ans). Yes, in the revised MS the parameter values and errors were mentioned up to two decimals.

  1. Line 457 - when expressing the MW values use the same precision throughout all manuscript - decide if using 2 or 3 decimals.

Ans). Thanks for grat suggestion, the entire MS was corrected and the given values precisely expressed up to two decimals.

  1. The Conclusions section has to be improved.

Ans). Thank for the valuable suggestion, the MS was revised and concluded with few millstones.